# Characteristics of Callus and Cell Suspension Cultures of Highbush Blueberry (*Vaccinium corymbosum* L.) Cultivated in the Presence of Different Concentrations of 2,4-D and BAP in a Nutrient Medium

**DOI:** 10.3390/plants13233279

**Published:** 2024-11-22

**Authors:** Dmitry A. Rybin, Alina A. Sukhova, Andrey A. Syomin, Tatiana A. Zdobnova, Ekaterina V. Berezina, Anna A. Brilkina

**Affiliations:** 1Department of Biochemistry and Biotechnology, Institute of Biology and Biomedicine, Lobachevsky State University of Nizhny Novgorod, Gagarin Avenue 23, 603022 Nizhny Novgorod, Russia; ivan.rybin.1990@mail.ru (D.A.R.); alinaismailova2000@bk.ru (A.A.S.); siom.andrei@yandex.ru (A.A.S.); berezina.kat@gmail.com (E.V.B.); 2Department of Biophysics, Institute of Biology and Biomedicine, Lobachevsky State University of Nizhny Novgorod, Gagarin Avenue 23, 603022 Nizhny Novgorod, Russia; t.zdobnova@mail.ru

**Keywords:** auxins, cytokinins, phenolic compounds, flavonoids, flavans, proanthocyanidins, productivity of cell cultures

## Abstract

In this work, cultures of callus and suspension cells originating from leaves of sterile highbush blueberry (*Vaccinium corymbosum* L.) plants were obtained and characterized. For their active growth and production of phenolic compounds, a combination of 2,4-D at a concentration of 0.34–2.25 µM and BAP at a concentration of 0.45–2.25 µM is effective. An increase in the phytohormone concentration leads to a slowdown in culture formation and reduces their ability to synthesize phenolic compounds. When cultivating *V. corymbosum* suspension cells over a year (12 passages), they not only retain the ability to synthesize phenolic compounds but also enhance it. By the 12th passage, the content of TSPC in suspension cells reaches 150 mg/g DW, the content of flavonoids reaches 100 mg/g DW, the content of flavans reaches 40 mg/g DW, and the content of proanthocyanidins reaches 30 mg/g DW. The high content of phenolic compounds may be due to the high expression of genes in flavonoid biosynthesis enzymes. *V. corymbosum* suspension cells accumulate a high level of phenolic compounds during a passage. The ability of *V. corymbosum* callus and cell suspension cultures in the presence of low concentrations of phytohormones to grow and accumulate biologically active phenolic compounds determines their high economic significance and prospects for organizing a biotechnological method for obtaining phenolic compounds.

## 1. Introduction

Callus and cell suspension cultures are an attractive source for the biotechnological production of valuable secondary plant metabolites, including a vast variety of phenolic compounds [1,2,3,4,5,6]. The main advantage of such cultures over plants from natural habitats is rapid biomass accumulation, regardless of the season, which opens up the possibility of scaling up the biotechnological process in small areas under strictly controlled conditions. To ensure unlimited proliferation of plant cells in vitro, exogenous phytohormones, such as auxins and cytokinins of natural or synthetic origin, are actively used in nutrient media composition. Most often, synthetic phytohormones are used to obtain and maintain callus and suspension cultures; these are auxin 2,4-dichlorophenoxyacetic acid (2,4-D) and cytokinin 6-benzylaminopurine (BAP; the synonym is 6-benzyladenine) [5,7,8,9,10,11].

2,4-D promotes cell proliferation to a greater extent than another synthetic auxin, α-naphthylacetic acid [12]. In addition, 2,4-D promotes an increase in vacuole size and, in general, the formation of looser callus tissues [13], which is important when subsequently obtaining cell suspension cultures. The effect of 2,4-D on plant cells is mediated primarily by the cytoplasmic receptor TIR1 (transport inhibitor response 1) [14,15], as well as the receptor on the plasma membrane and endoplasmic reticulum ABP1 (auxin-binding protein 1) [16]. The signaling pathway through the TIR1 receptor leads to cell division and differentiation and possesses low affinity for auxin [17], while the signaling pathway through the ABP1 receptor leads to cell elongation and stretching and possesses higher affinity for auxin [18]. BAP binds with high affinity to the receptors CRE1/AHK4 (cytokinin response1/arabidosis histidine kinase4), AHK2, and AHK3 [19]. Auxins and cytokinins play an important role in the cell cycle: auxins act as a permissive signal for cell division, providing the G1/S transition [14], and cytokinins provide the G1/S and G2/M transitions [20]. Auxins can control cytokinin biosynthesis through the specific activation of isopentenyl transferases IPT5 and IPT7 [21], and cytokinins can inhibit indole-3-acetic acid oxidation [22]. Moreover, auxins and cytokinins mutually regulate their signaling factors and/or their metabolism [20]. However, high concentrations of auxins and cytokinins inhibit the growth of cell and tissue cultures [23,24]. 

It is known that auxins and cytokinins enhance the expression of genes of phenolic compound biosynthesis enzymes: phenylalanine ammonia lyase (PAL), cinnamic acid 4-hydroxylase (C4H), chalcone synthase (CHS), chalcone isomerase (CHI), flavanone-3-hydroxylase (F3H), flavonol synthase (FLS), dihydroflavonol-4-reductase (DFR), as well as a number of regulatory genes [25,26,27,28]. At the same time, there is information about the negative effect of auxins on the accumulation of phenolic compounds, in particular, anthocyanins [29,30]. The inconsistency of such information may be associated, among others, with the concentrations of the phytohormones used [26,30]. 

When working with callus and cell suspension cultures, phytohormone concentrations of up to 10 mg/L are used, most often 0.1–5 mg/L [5,8,10]. Considering that such cultures are used to obtain valuable secondary metabolites, including phenolic compounds, it is necessary to select the most effective concentrations and combinations of phytohormones for each specific case of cultivation. 

Highbush blueberry (*Vaccinium corymbosum* L., Ericaceae Juss.) accumulates a great many phenolic compounds in its tissues [11,31,32]. Extracts from its berries and leaves are actively studied due to the possibility of using them as a promising adaptogenic and therapeutic agent for a wide range of pathological conditions, including liver damage, cardiovascular disease, neurodegenerative disorders, diabetes, obesity, etc. [33,34,35,36]. Therefore, given its wide application, *V. corymbosum* can be used to obtain phenolic compounds from cell cultures along with such well-known phenol producers as *Lithospermum erythrorhizon*, *Lavandula vera*, *Ocimum basilicum* [37].

There are several works on obtaining callus cultures from different parts of *V. corymbosum* plants. In particular, calluses were obtained from leaves, stems, and roots [5,8,10] using auxins and cytokinins at a concentration of up to 5 mg/L. For the initiation of *V. corymbosum* calluses from leaves and berries of plants from greenhouses, a phytohormone 2,4-D/BAP combination was used at a concentration of 0.5/0.5 mg/L [11]. Later, using the same phytohormone combination, cell suspension cultures were successfully obtained from the calluses [11]. These calluses, and especially cell suspension cultures of *V. corymbosum,* were characterized by a high content of total soluble phenolic compounds (TSPCs), flavonoids (including flavans), and proanthocyanidins, which exceeded berries, and, in some cases, leaves [11]. However, the effect of other concentrations of 2,4-D and BAP on biomass and phenolic compound accumulation during one passage and over the long-term passaging of callus and cell suspension cultures, as well as on the expression of genes of phenolic compound biosynthesis enzymes, remains unclear. In this work, we undertake an investigation that can shed light on the questions posed above, namely, to identify the effect of 2,4-D and BAP concentrations in a range from 0.34 to 6.75 μM (0.075–1.5 mg/L) on biomass and phenolic compound accumulation in calluses and on biomass and phenolic compound accumulation in suspension cells of *V. corymbosum* during one passage and over the long-term passaging and on the expression of genes of phenolic compound biosynthesis enzymes.

## 2. Results

### 2.1. Morphology and Biomass of V. corymbosum Calluses

*V. corymbosum* calluses were obtained from leaves of sterile plants on a solid WPM nutrient medium with different concentrations of 2,4-D and BAP, varying in appearance, size, and weight (Table 1). The largest and, at the same time, friable calluses were obtained in the presence of 2,4-D/BAP in concentrations of no more than 1.7/2.25 µM. An increase in the phytohormone concentration led to a decrease in the callus size, and calluses appeared compact. Calluses reached significant fresh and dry weights (over 0.15 and 0.015 g per callus, respectively) on media with the addition of no more than 3.4 µM 2,4-D and 4.5 µM BAP. When using higher concentrations of phytohormones, the callus weight decreased almost threefold. Also, an increase in the phytohormone concentration led to callus browning.

### 2.2. Phenolic Compound Accumulation in V. corymbosum Calluses 

The presence of different concentrations of 2,4-D and BAP in nutrient media also affected the phenolic compound content in *V. corymbosum* calluses (Figure 1). An increase in the phytohormone concentration led to a decrease in the level of TSPCs, flavonoids, and flavans. The highest TSPC content (more than 50 mg/g DW) was observed in calluses obtained on a nutrient medium with 2,4-D/BAP at concentrations of 0.34/0.45, 1.7/2.25, and 3.4/4.5 µM. Flavonoids, including flavans, make up a significant part of the callus phenolic complex. The highest flavonoids content was found when using 0.34/0.45 and 0.45/0.45 µM 2,4-D/BAP, and the highest flavans content was found when using 0.45/0.45 µM 2,4-D/BAP. All phytohormone concentrations did not affect proanthocyanidin accumulation (Figure 1).

### 2.3. Characteristics of V. corymbosum Cell Suspension Cultures 

*V. corymbosum* calluses obtained in the previous stage of this study were used to obtain cell suspension cultures. For this purpose, calluses cultivated on a solid nutrient medium with a certaat a concentration of 2,4-D and BAP were transferred to a liquid nutrient medium with the same phytohormone concentrations. After four passages, cell suspension cultures were formed (Table 2). These cultures were formed of small cell aggregates and individual cells. The efficiency of the cultures forming depended on 2,4-D and BAP concentrations in nutrient media. 2,4-D at a concentration of 0.34–3.4 µM and BAP at a concentration of 0.45–4.5 µM turned out to be effective. In the case of using 2,4-D and BAP at a concentration of 4.5/4.5 µM and higher, cell suspension cultures were not formed. The highest cell count (up to 22.8 million cells/L) and viability (84%) after four passages were observed in the culture initiated using 0.45/0.45 µM 2,4-D/BAP. During further subcultivation, by the 12th passage, the cultures reached a high density of 25–45 million cells/L. The most intensive growth was observed for the culture where the lowest concentration of phytohormones—0.34/0.45 µM 2,4-D/BAP—was used. In the case of using 3.4/4.5 µM 2,4-D/BAP, cell growth was inhibited. 

In the suspension cultures, cell weight changed in a similar way (Figure 2). After the fourth passage, the fresh and dry weights of cells cultivated in the presence of 4.5/4.5 µM 2,4-D/BAP were significantly lower than the cell weights of the remaining cultures. For suspension cells cultivated in the presence of 0.34/0.45–2.25/2.25 µM 2,4-D/BAP, an increase in both fresh and dry weights was observed from the 4th to the 12th passage. The highest fresh and dry cell weights after the 12th passage were observed in the presence of 0.34/0.45 and 0.45/0.45 µM 2,4-D/BAP (Figure 2).

### 2.4. Phenolic Compound Accumulation in V. corymbosum Cell Suspension Cultures 

*V. corymbosum* suspension cells were characterized by a significant accumulation of phenolic compounds (Figure 3). In young cultures after the fourth passage, the TSPC level in cells when using 0.34/0.45–1.7/2.25 µM 2,4-D/BAP was similar and reached 80–100 mg/g DW. In addition, the levels of flavonoids, flavans, and proanthocyanidins when using these phytohormone concentrations were similar, too. However, the use of 2.25/2.25 µM 2,4-D/BAP caused a sharp decrease in the content of all groups of phenolic compounds in suspension cells. 

In well-formed *V. corymbosum* cell suspension cultures, with further subcultivation, the content of TSPC, favonoids, flavans, and proanthocyanidins increased (Figure 3). After the 12th passage, the highest content of TSPC, flavans, and proanthocyanidins was found in suspension cells cultivated with the addition of 0.34/0.45 and 0.45/0.45 µM 2,4-D/BAP. For example, the TSPC content reached 150 mg/g DW. At the same time, the flavonoids content (110 mg/g DW) was the highest in suspension cells cultivated with the addition of 2.25/2.25 µM 2,4-D/BAP. It should be noted that the levels of TSPC, flavans, and proanthocyanidins in these suspension cells were also quite high. In general, with an increase in the cultivation time of *V. corymbosum* suspension cells, the phenolic compound accumulation increased with the use of 2,4-D and BAP in a concentration range from 0.34/0.45 to 2.25/2.25 µM (Figure 3). 

### 2.5. Expression of Genes of Proteins Involved in Flavonoids Biosynthesis 

In *V. corymbosum* suspension cells in the presence of 2.25/2.25 μM 2,4-D/BAP, after the 12th passage, the increase in the phenolic compound content was the most significant (in particular, content of flavonoids and proanthocyanidins increased by 10-times; Figure 3). The use of 0.34/0.45 μM 2,4-D/BAP, with a general positive effect, did not increase only the flavonoids content. To understand the reasons for such a difference in flavonoids content in suspension cells grown in the presence of 2,4-D/BAP in concentrations of 0.34/0.45 and 2.25/2.25 μM, the expression of a number of genes encoding flavonoid biosynthesis enzymes was assessed: chalcone synthase (CHS), chalcone isomerase (CHI), flavanone-3β-hydroxylase (FHT), dihydroflavonol reductase (DFR), leucoanthocyanidin reductase (LAR), as well as the transcription factor MYBPA1. 

When cultivating suspension cells in the presence of a higher (2.25/2.25 µM) concentration of 2,4-D/BAP, the relative expression levels of all the studied genes, except CHS, were significantly higher than in the presence of a lower (0.34/0.45 µM) concentration of 2,4-D/BAP (Figure 4). The expression of FHT increased especially noticeably with an increase in phytohormone concentration, approximately 7.8-times, while the expression of other genes increased 2–3-times.

### 2.6. Growth and Phenolic Compound Accumulation in V. corymbosum Suspension Cells During One Passage

*V. corymbosum* suspension cells cultivated in the presence of 0.34/0.45 µM 2,4-D/BAP were characterized by intensive growth during passage. Suspension cell growth was accompanied by the presence of a 6–9-day lag phase (Figure 5A). The stationary phase occurred on days 27–30. This cell suspension culture can be considered to grow well, since the growth index (I) calculated for fresh biomass was 13.32, the specific growth rate (µ) was 0.13 day^–1^, the biomass doubling time (τ) was 5.31, the economic coefficient (Y) was 0.297, the biomass productivity (P) was 2.97 g/L per day (Figure 5B) (dry biomass productivity was 0.304 g/L per day), the maximum fresh biomass accumulation was 98.6 g/L, and the maximum dry biomass accumulation was 10.5 g/L.

The analysis of phenolic compound accumulation in this cell suspension culture was performed on the 12th, 18th, 24th, and 30th days. The maximum level of TSPCs, flavonoids, and flavans occurred on the 18th day, in the middle of the exponential growth phase (Figure 5C). Then, in the growth slowdown phase (the 24th day), the phenolic compound level decreased, and the flavonoids level continued to decrease until the end of the passage. The proanthocyanidins content was high both in the exponential growth phase and in the stationary growth phase.

## 3. Discussion

The co-presence of auxins and cytokinins is necessary for the preparation of nutrient media for calluses and suspension cell cultivation. The phytohormones effect on the morphological and biochemical characteristics of cultures depends on the concentration used. The most commonly used concentrations of auxins and cytokinins, including 2,4-D and BAP, are from 0.1 to 5 mg/L (about 0.45–22.5 μM). For example, for *Cnidium officinale* calluses, the most effective concentrations of phytohormones were 2.3 μM 2,4-D and 2.2 μM BAP [7] and for *Ficus religiosa* calluses, 0.5 mg/L 2,4-D and 0.05 mg/L BAP [38]. The fastest growth of *Garcinia mangostana* L. suspension cells was observed on a medium containing 2.26 μM 2,4-D and 2.22 μM BAP [39].

We have shown that large, soft, and loose light-colored *V. corymbosum* calluses were formed when using 2,4-D and BAP at a concentration of no more than 2.25 μM (Table 1). An increase in the phytohormone concentration led to a decrease in the size and weight of initiated calluses and to the acquisition of a denser structure and dark (brown) color. Similar results were obtained on *Cnidium officinale* calluses when using 2,4-D at a concentration of up to 6.8 μM, as well as when using 2,4-D at a concentration of up to 6.8 μM together with BAP at a concentration of 2.2 μM [7]; *Clinacanthus nutans* calluses when using 2,4-D and BAP at a concentration of up to 0.5/1 mg/L (2.25/4.5 μM) [9].

High values of accumulated biomass indicate such cultivation conditions under which cells will probably retain the ability to undergo long-term passaging. On the contrary, low values of accumulated biomass, as well as cell browning (a typical problem of calluses of woody plants and plants rich in secondary metabolites), indicate a decrease in synthetic processes and aging of the culture. The aging of plant cells, suppression of their division, compaction due to the formation of secondary cell wall and its lignification can be associated with the auxin-induced accumulation of ethylene [9]. In the case under consideration, such an effect of auxin 2,4-D is clearly dose-dependent and manifests itself with an increase in the 2,4-D concentration of above 2.25–3.4 μM. With an increase in the phytohormone concentration, compact calluses are formed, as we have shown for *V. corymbosum* calluses. In addition, this was shown for *Amorphophallus muelleri* [40] and *Catharanthus roseus* [41] calluses. Ethylene also has a negative effect on the accumulation of biomass and secondary metabolites, namely carotenoids, anthocyanins, and proanthocyanidins, in suspension cells, as shown for *V. pahalae* [42]. 

For *Arabidopsis thaliana* suspension cells, it was shown that high concentrations of BAP promote the development of oxidative processes, early expression of SAG12, a cysteine protease involved in programmed cell death, which also negatively affects the biometric characteristics of calluses [43]. In the case of the development of oxidative processes, phenolic compounds can be oxidized to quinones under the action of polyphenol oxidase, the activity of which is increased, for example, in brown calluses of *Ranunculus asiaticus* [44] and *Pinus sylvestris* [45]. Quinones, together with tannins and melanins, can lead to *P. sylvestris* calluses browning and inhibit the activity of a number of enzymes: peroxidase, carbohydrate metabolism enzymes invertase and sucrose synthase, etc. [45]. A decrease in the activity of such enzymes should also negatively affect the biometric characteristics of calluses. At the same time, a decrease in the expression level and activity of polyphenol oxidase reduces the ability of calluses to oxidize phenolic compounds and to acquire a brown color, which was demonstrated for transgenic *Malus* × *domestica* calluses [46].

In brown *V. corymbosum* calluses initiated using elevated concentrations of 2,4-D and BAP (4.5 μM and higher), we noted a decrease in the content of TSPCs, flavonoids, and flavans (Figure 1). At the same time, the content of oligomeric phenolic compounds—proanthocyanidins—did not differ from that in calluses initiated using lower concentrations of 2,4-D and BAP. Accordingly, when using elevated concentrations of 2,4-D and BAP (4.5 μM and higher), the proanthocyanidin representation in the phenolic complex of *V. corymbosum* calluses increased. The participation of proanthocyanidins in callus browning was discussed in relation to *Camellia hainanica* species [47]. In brown undifferentiated calluses of *R. asiaticus*, polymeric phenolic compounds, the content of which was high (compared to differentiated calluses), could contribute to callus browning, while the content of a number of phenolic acids and flavonoids, on the contrary, was low [44]. We assume that for brown undifferentiated *V. corymbosum* calluses, initiated using elevated concentrations of 2,4-D and BAP (4.5 μM and more), an increase in the content of and/or representation of polymeric phenolic compounds, i.e., tannins and melanins, cannot be excluded; however, we did not determine their content and the content of quinones. 

It should be emphasized that in *V. corymbosum* calluses cultivated in the presence of low concentrations of 2,4-D and BAP (up to 4.5 μM), the content of TSPCs, flavonoids, and flavans was 1.5–2-times higher than in the presence of elevated concentrations of 2,4-D and BAP, comparable with calluses of *V. macrocarpon*, *V. oxycoccos* [3] and *V. corymbosum* cv. Bluecrop and Duke [8]. Together with the preferable morphometric characteristics of *V. corymbosum* calluses (loose structure, light color, high biomass accumulation), the increased biosynthetic activity of calluses initiated using 2,4-D and BAP concentrations of no more than 2.25 μM makes them more economically viable objects for further obtaining cell suspension cultures. 

To initiate *V. corymbosum* cell suspension cultures, we used the same concentrations of phytohormones as for callus initiation in order to assess whether the effects of different concentrations of 2,4-D and BAP that we identified would be similar in relation to an object with another organization, i.e., in relation to individual cells suspended in a liquid nutrient medium. Our assumption about the preferable use of 2,4-D and BAP at concentrations of no more than 2.25 μM for the initiation of *V. corymbosum* cell suspension cultures was justified. Calluses obtained using these concentrations turned out to be a productive inoculum for the formation of cell suspension cultures with a high degree of disaggregation (in comparison with variants using high concentrations of 2,4-D and BAP), viability (Table 2), and the ability to accumulate biomass (Figure 2) and phenolic compounds (Figure 3). And, though information on the effect of different phytohormone concentrations on callus formation for different plant species is presented quite widely in the literature [7,9,10], information on their effect on cell suspension culture formation is almost never presented; therefore, our results on the effect of different concentrations of 2,4-D and BAP on *V. corymbosum* cell suspension culture formation from calluses are novel and relevant. 

When subculturing *V. corymbosum* suspension cells in the presence of low concentrations of 2,4-D and BAP (no more than 2.25 μM), we noted an improvement in the morphometric (cell number, viability, and biomass) and biosynthetic (accumulation of phenolic compounds) characteristics after the 12th passage compared to those after the fourth passage. An improvement in biosynthetic characteristics was also reported for anthocyanin-accumulating suspension cells of *V. ashei* and *Cleome rosea* when they were subcultivated in media supplemented with only 2,4-D at a concentration of no more than 0.45 μM for 57 passages [48] and for 4 passages [49], respectively. For *C. rosea*, an increase in biomass accumulation did not occur [49], and, therefore, the feature of long-term subcultivated *V. corymbosum* suspension cells that we identified can be explained by the presence of cytokinin BAP in the nutrient medium and/or growth-stimulating activity of some non-anthocyanin phenolic compounds. Clarification of this issue should be of great fundamental importance for understanding the role of phytohormones and secondary metabolites in plant cells, as well as of great practical importance for ensuring the economic efficiency of biotechnological production of plant secondary metabolites.

Similar levels of phenolic compounds in suspension cells have been revealed in a number of studies. Thus, in cell suspension cultures of *Buddleja cordata,* the TSPC level reached 152 mg/g DW [50]; *Ageratina pichichensis*—91 mg/g DW [51]; *Lycium schweinfurthii*—84 mg/g DW (and flavonoids level reached 19 mg/g DW) [52]; *Dendrobium fimbriatum*—51 mg/g DW [53]. But, more often, the level of phenolic compounds in cell suspension cultures is quite low and ranges from 5 to 24 mg/g DW for TSPC and 3.7–8 mg/g DW for flavonoids [53,54,55,56]. It should also be noted that, unfortunately, there is practically no information on the content of TSPCs and flavonoids in cell suspension cultures of plants of Vaccinium L. genus. In earlier studies, for example, devoted to *V. ashei* and *V. pahalae* cell suspension cultures, only the content of anthocyanins and proanthocyanidins or individual phenolic compounds was assessed [48,57]. In our previous work devoted to *V. corymbosum* cell suspension cultures obtained from leaves and berries of plants from a greenhouse, the level of phenolic compounds [11] was similar to that presented in this work. Based on that, it can be concluded that *V. corymbosum* cell suspension cultures, accumulating TSPC up to 150 mg/g DW and flavonoids up to 100 mg/g DW, are valuable phenolic compounds synthesizing cultures. 

Flavonoids are one of the most spread groups of phenolic compounds, and their synthesis pathway is the most branched and diverse. It has been shown that the regulation of flavonoid metabolism occurs largely at the transcriptional level with transcription factors belonging to the MYB, bHLH, and WD40 protein families and functioning as a single MBW complex [27,29]. Hormone-sensitive elements have been identified in gene promoters of both phenolic compound biosynthesis enzymes and their transcription factors [26,58,59]. To identify features of the regulation of expression of phenolic biosynthesis genes in *V. corymbosum* suspension cells and to identify key points of the flavonoid biosynthesis scheme that are sensitive to the phytohormone concentration, an analysis of gene expression was performed at the level of mRNA of sequentially working enzymes: CHS—a switching enzyme of flavonoid biosynthesis, providing synthesis of the first flavonoid—narigenin chalcone (chalcone), CHI—of naringenin (flavanone), FHT—of dihydrokaempferol (dihydroflavonol), DFR—of leucoanthocyanidins, LAR—of flavan-3-ols (from leucocyanidins), as well as expression of gene of transcription factor MYBPA1, which activates the expression of *DFR* and *LAR* genes [60]. 

An increase in the expression levels of the *CHI*, *FHT*, *DFR*, *LAR*, and *MYBPA1* genes with an increase in 2,4-D and BAP concentration indicates that the activation of flavonoid biosynthesis in response to these phytohormones affects all key stages of flavonoid biosynthesis, both early and late. The absence of a significant change in the expression level of the *CHS* gene may be because of the chosen time point for expression analysis, conducted at the end of the passage, i.e., 30 days after transplanting cells in a fresh nutrient medium. Perhaps a noticeable activation of *CHS* gene expression could occur earlier, and, at the chosen time point, its expression level was maintained at a certain “basic” level, and differences in flavonoids accumulation are due to activation at further stages of the flavonoid pathway. Previously, the level of *CHS*, *FHT*, *DFR*, *LAR*, and *MYBPA1* gene expression was determined in *V. corymbosum* plants, mainly in flowers and berries [29,61,62]. Here, we report the level of *CHS*, *CHI*, *FHT*, *DFR*, *LAR*, and *MYBPA1* gene expression in *V. corymbosum* suspension cells for the first time. 

To assess the growth of *V. corymbosum* suspension cells and phenolic compound accumulation during the passage, we evaluated the biomass growth curve and the nature of accumulation of phenolic compounds by suspension cells cultivated in the presence of the lowest concentrations of 2,4-D/BAP (0.34/0.45 µM) used. The growth curve has a typical S-shape (Figure 5A). The lag phase, characterized by the absence of cell growth and division, was about 9 days. Such a long lag phase may be a consequence of the low initial density of suspension cells and their slow adaptation to the fresh nutrient medium. Upon completion of adaptation to the medium, the cells began to actively absorb nutrients and divide, and the exponential phase in *V. corymbosum* suspension cells lasted up to the 27th–30th day. A sufficiently high economic coefficient Y (0.297; Figure 5B) shows that almost 30% of nutrient medium sucrose is spent on building cell biomass. A similar growth cycle is characteristic of cell suspension cultures of *Artemisia absinthium* (lag phase—6 days, stationary phase—starting from the 27th day) [63], *Jatropha curcas* (lag phase—7 days, stationary phase—starting from the 35th day) [64], *Digitalis lanata* (lag phase—6–8 days, stationary phase—starting from the 28th day) [65], and the cell suspension culture of *Cajanus cajan* has even longer growth cycle phases (lag phase—14 days, stationary phase—starting from the 49th day) [66]. Similar I, Y, and P growth parameters (Figure 5B) and productivity are characteristic of cell suspension cultures of *Phlojodicarpus sibiricus* [4], *D. lanata* [65], *Dioscorea deltoidea*, and *Panax japonicus* [67]. The cultures of *D. deltoidea* and *P. japonicus* [67] have the same µ value as *V. corymbosum*, but the abovementioned cultures of *P. sibiricus* [4], *D. lanata* [65], as well as several other cultures, such as *J. curcas* [64], *A. pichinchensis* [68], *Eysenhardtia platycarpa* [69], for which growth parameters and growth curves are given in the literature, µ is almost twice as large, and τ, as a consequence, is almost half as small. The differences are due to slightly shorter growth cycles and a higher level of biomass accumulation than in our case. It is noteworthy that the content of TSPCs and flavonoids in our cell suspension culture of *V. corymbosum* is one to two orders of magnitude higher than in the cell suspension culture of *E. platycarpa*, which is characterized by a shorter growth cycle, higher µ value, and lower τ value [69]. 

As for the dynamics of phenolic compound accumulation in *V. corymbosum* suspension cells during passage, the level of phenolic compounds in the middle of the exponential phase (the 18th day) increased by about 30% compared to their level in cells in the early exponential phase (the 12th day). And, then, in the growth slowdown phase (the 24th day), it decreased by 35–40%, after which, by the stationary phase (the 30th day), it increased again; however, for different groups of phenolic compounds, this second increase was different. Thus, the proanthocyanidin level on the 30th day became equal to that on the 18th day; the flavans level to that on the 12th day; the TSPC level was intermediate between the 12th and the 18th days. On the contrary, the flavonoids level decreased. The decrease in the content of flavonoids against the increase in the content of the other phenolic compounds studied can be explained by flavonoid oligomerization with proanthocyanidins formation, since it was the level of proanthocyanidins that increased most significantly in the period between the 24th and the 30th days of passage. This assumption, however, requires confirmation using biochemical (determination of qualitative and quantitative composition of individual phenolic compounds) and molecular (determination of the level of expression of genes of phenolic biosynthesis enzymes) approaches. Nevertheless, it is obvious that suspension cells cultivated in the presence of 0.34/0.45 μM 2,4-D/BAP accumulate phenolic compounds most intensively on the 18th day of passage, and by the 30th day, their content may decrease somewhat. 

A comparison of the results on biomass accumulation and on phenolic compound accumulation during passage and at the end of different passages (Figure 3 and Figure 5) allows us to conclude that *V. corymbosum* suspension cells are characterized by significant synthesis of phenolic compounds during the period of cell biomass accumulation. The coincidence of the period of phenolic compound accumulation with the period of biomass accumulation was also reported for the cell suspension culture of *C. rosea*, in which the maximum accumulation of biomass and anthocyanins occurred on the 14th–18th days of passage [49].

Thus, we obtained and characterized *V. corymbosum* callus and cell suspension cultures, which are active producers of phenolic compounds. It was revealed that the most intensive growth and phenolic compound (TSPC, flavonoids, flavans, and proanthocyanidins) accumulation occur when using 0.34–2.25 µM 2,4-D and 0.45–2.25 µM BAP. The prospect of this study will be an investigation of individual phenolic compound representation in the phenolic complex and the regulation of their synthesis.

## 4. Materials and Methods

### 4.1. Plant Material, Initiation of Callus and Cell Suspension Cultures

We used highbush blueberry plants (*Vaccinium corymbosum* L.) cultivated in vitro on agarized Woody Plant Medium (WPM) without phytohormones, containing 3% sucrose and 0.8% agar, pH 5.0 before autoclaving [70]. Calluses were initiated from leaves of these plants by placing leaves in 100 mL culture vessels with 20 mL of WPM nutrient medium supplemented with 2,4-D and BAP in concentrations of 0.34/0.45 (0.075/0.1); 0.45/0.45 (0.1/0.1); 1.7/2.25 (0.375/0.5); 2.25/2.25 (0.5/0.5); 3.4/4.5 (0.75/1); 4.5/4.5 (1/1); 5/6.75 (1.25/1.5); 6.75/6.75 (1.5/1.5) µM (mg/L). Callus initiation lasted for 8 weeks. Then, after one cultivation cycle (passage) of 4 weeks, cell suspension cultures were initiated from callus cultures in the same nutrient media but without addition of agar. For this purpose, calluses weighing about 1 g were transferred to 100 mL flasks containing 30 mL of WPM nutrient medium and cultivated on an orbital shaker at a rotation speed of 120 rpm with a rotation radius of 20 mm. To obtain suspension cells, calluses initiated on all variants of nutrient media with all concentrations of phytohormones were used. The duration of passages was 30 days. For the first three passages, the inoculum/fresh medium ratio during transplantation of cell suspension culture was 1:2; subsequently, the ratio was reduced to 1:6. Calluses and suspension cells were cultivated in the dark at 25 °C and 60–70% air humidity. Callus and cell suspension cultures were initiated 5 times. In each experiment, at least 30 calluses and 10 flasks with suspension cells were obtained for each variant of the nutrient medium.

### 4.2. Determination of Growth Parameters of Callus and Cell Suspension Cultures

To assess the development of callus and cell suspension cultures after the end of cultivation cycles (passages), their fresh and dry weights were determined. In the case of calluses, fresh and dry weights of one callus were determined. In the case of suspension cells, fresh and dry weights were calculated per 1 L of nutrient medium, for which an aliquot of cells in nutrient medium (30 mL) was filtered on a Buchner funnel through a filter paper under vacuum, transferred to a weighing cup, and weight was determined. To determine dry weight, calluses and suspension cells were dried at 60 °C to constant weight.

Callus size was estimated by analyzing callus images in photographs using the ImageJ program. A centimeter ruler was used as a standard. Callus size was expressed in cm^2^.

Suspension cells were counted using a Nageotte counting chamber and an MT5300L/SP microscope (Meiji Techno, Saitama, Japan), a UPlan 10×/0.25 objective, and a ToupCam digital camera (ToupTek, Hangzhou, China). The viability of suspension cells was determined by staining them with a 0.025% solution of the vital dye Evans blue (Sigma, St. Louis, MI, USA).

For suspension cells cultivated in a nutrient medium with a minimal concentration of phytohormones (0.34/0.45 μM 2.4-D/BAP), growth curve was estimated during one passage. The growth curve was plotted based on cell fresh and dry weight measured every 3 days for 30 days. Also, on the 12th, 18th, 24th, and 30th days, content of phenolic compounds in suspension cells was determined. Growth parameters of suspension cells were determined by calculating growth index (I), specific growth rate (μ), biomass doubling time (τ), economic coefficient (Y), and biomass productivity (P). The following formulas were used for calculations [65]:

I = X_max_/X_0_, where X_max_ and X_0_ are the maximum and initial values of cell fresh weight, respectively.

µ = (lnX_2_ − lnX_1_)/(t_2_ − t_1_), where X_2_ and X_1_ are the values of cell fresh weight at time t_2_ and t_1_, respectively.

τ = ln2/µ.

P = (X_i_ − X_0_)/(t_i_ − t_0_), where X_0_ and X_i_ are the values of cell fresh weight at the beginning of cultivation and at time t_i_, respectively. 

Y = (X_max_ − X_0_)/S_0_, where X_max_ and X_0_ are the maximum and initial concentrations of dry cells biomass (g/L), respectively; S_0_ is the initial concentration of the substrate (sucrose) in the medium (g/L of medium).

### 4.3. Extraction and Analysis of Phenolic Compound Accumulation in Callus and Cell Suspension Cultures

*V. corymbosum* calluses and suspension cells were fixed with boiling 80% ethanol solution for 10 min [3]. In the obtained extracts, content of total soluble phenolic compounds (TSPCs) and of flavans was determined according to M.N. Zaprometov [3], content of flavonoids was determined according to [71], and content of proanthocyanidins was determined according to [72]. All measurements were conducted colorimetrically using an SF-2000 spectrophotometer (OKB Spektr, Saint Petersburg, Russia).

For TSPCs, 0.5 mL of extract, 8.55 mL of water, 0.5 mL of Folin-Denis reagent, and 1 mL of 7% Na_2_CO_3_ solution were mixed, and after 1 h, optical density of the obtained mixture was measured at a wavelength of 725 nm.

For flavonoids, 0.5 mL of extract, 6.9 mL of water, 0.3 mL of 5% NaNO_2_ solution, 0.3 mL of 10% AlCl_3_ solution, 2 mL of 1 M NaOH solution were mixed, and optical density of the obtained mixture was measured at a wavelength of 510 nm.

Rutin (Acros Organics, Geel, Belgium) stock solution (10 mg per 50 mL of 50% ethanol solution) was used as a standard for calculation of TSPC and flavonoids content.

For flavans, 0.4 mL of extract and 2 mL of vanillin reagent (0.1% vanillin solution in HCl_conc._) were mixed, and after 15 min, optical density of the obtained mixture was measured at a wavelength of 510 nm. Catechin (Sigma, St. Louis, MI, USA) stock solution (10 mg per 50 mL of 50% ethanol solution) was used as a standard for calculation of flavans content.

For proanthocyanidins, 0.4 mL of extract, 2.4 mL of butanol reagent (butanol/HCl_conc._ = 19:1) and 0.12 mL of Fe(III) reagent (0.5 g of Mohr’s salt in 25 mL of 2 M HCl solution) were mixed. The obtained mixture was heated in a water bath at 95 °C for 40 min. Optical density of the mixture was measured at a wavelength of 550 nm. The following formula was used for calculation:$$C\;=\;D\;\times \frac{\mathrm{Mr}}{\varepsilon \;\times \;l}\times \;F,\;$$
where D is optical density; Mr is cyanidin molar mass (287 g/mol); ε is coefficient of cyanidin molar extinction (34,700 M^–1^·cm^–1^) [73]; l is optical path length, cm; F is dilution factor. 

All measurements were performed in three biological replicates. Each biological replicate comprises three analytical replicates. The results were expressed in mg/g DW.

### 4.4. RNA Isolation and cDNA Synthesis

Total RNA was isolated from *V. corymbosum* suspension cells by a modification of the method described in [61]. Around 120–140 mg of cells was ground in liquid nitrogen, transferred to a CTAB buffer (2% CTAB, 2% PVP K30, 100 mM Tris-HCl, pH 8.0, 25 mM EDTA, 2 M NaCl, 0.5 g/L spermidine, 2% β-mercaptoethanol) heated to 65 °C and incubated for 45 min at 65 °C. The resulting extract was then centrifuged for 10 min at 14,000× *g* to remove residual plant tissue. An equal volume of chloroform was added to the supernatant, mixed and centrifuged at 12,000× *g* for 15 min at 4 °C. The chloroform extraction was repeated twice. The upper aqueous layer was collected in a new 1.5 mL tube, and ¼ volume of 10 M lithium chloride was added, mixed and incubated overnight at 4 °C. Then, the mixture was centrifuged for 25 min at 12,000× *g* at 4 °C. The supernatant was decanted, the precipitate was dissolved in 20 μL of mQ water, and 2 μL of 3 M sodium acetate and 55 μL of ethyl alcohol were added, mixed, and incubated for 1 h at –20 °C. Then, the mixture was centrifuged for 1 h at 14,000× *g* at 4 °C. The resulting precipitate was washed twice with ethanol, dried and dissolved in 20 μL of mQ water. The quantity and quality of the obtained RNA were determined using a NanoVue drop spectrophotometer (GE Healthcare, Chicago, IL, USA). The integrity of the obtained RNA was assessed by electrophoresis in 1% agarose gel. 

To avoid possible contamination with genomic DNA, the isolated RNA was treated with DNAse I, RNAse-free (Thermo Scientific, Waltham, MA, USA). The obtained RNA was subjected to reverse-transcription reaction using MMLV RT kit with Oligo(dT)-primer (Evrogen, Moscow, Russia).

### 4.5. Real-Time PCR

Relative gene expression was assessed by real-time PCR (rtPCR) using a 7500 Real-Time PCR System (Applied Biosystems, Foster City, CA, USA), qPCRmix-HS SYBR + HighROX kit (Evrogen, Moscow, Russia) and primers shown in Appendix A. The PCR program was as follows: (1) initiation at 50.0 °C for 2 min, (2) “hot start” at 95.0 °C for 10 min, (3) denaturation at 95.0 °C for 15 s, (4) primer annealing and DNA synthesis at 60.0 °C for 1 min. Stages 3–4 were repeated for 40 cycles. Determination of Ct (threshold cycle) was performed with 7500 Software v. 2.0.4 (Applied Biosystems, Foster City, CA, USA).

The data from the rtPCR experiment were presented as 2^–∆Ct^ (as described for individual data point in [74]), where ∆Ct = Ct (gene of interest) − Ct (reference gene). *UBQ3b*, *GAPDH*, *EF1α* were used as reference genes; the data were presented as geometric mean of three reference genes The experiment was performed in four biological replicates.

### 4.6. Statistical Analysis

One biological replicate included three flasks with suspension cells and 5–7 calluses. The graphs present arithmetic mean values from at least three biological replicates and their standard deviations. Statistical processing of results was performed using GraphPad Prism 9 software (GraphPad Software, San Diego, CA, USA). The reliability of the values was determined using Tukey criterion and in the case of gene expression analysis, using unpaired *t*-test. Differences were considered statistically significant at *p* < 0.05.

## 5. Conclusions

In this work, we found that phytohormones 2,4-D and BAP in concentrations of no more than 2.25 µM in nutrient media provide an accumulation of biomass and significant amount of phenolic compounds in both the *V. corymbosum* callus and cell suspension cultures. An increase at a concentration of these phytohormones leads to a slowdown in growth, cell browning, decrease in viability, and inability to synthesize phenolic compounds. In cell suspension cultures, it is also accompanied by the formation of large aggregates that interfere with cell subcultivation. Possibly, in vitro cultivated *V. corymbosum* cells are characterized by relatively high levels of endogenous auxins and cytokinins or high receptor sensitivity to 2,4-D and BAP. This issue had not been studied until now, and our study emphasizes the importance and prospects of clarifying it. Our study demonstrates the possibility of using exogenous auxins and cytokinins in the lowest concentrations from a range of 0.1–5 mg/L, in other words, the possibility of reducing the cost of nutrient media. This undoubtedly has high economic importance and makes the *V. corymbosum* callus and cell suspension cultures a promising object for the organization of a biotechnological method for phenolic compound production.

## Figures and Tables

**Figure 1 plants-13-03279-f001:**
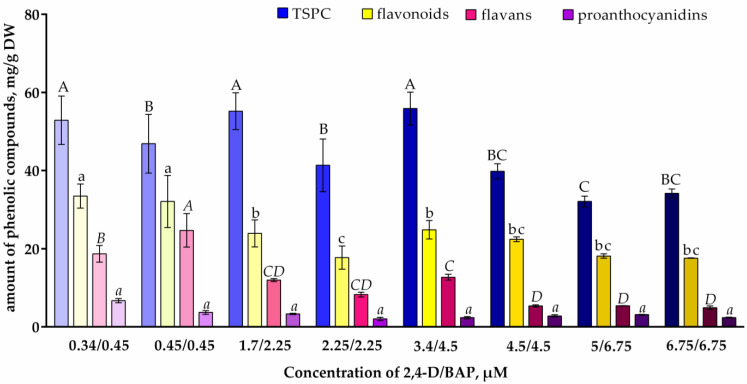
Phenolic compounds content in *V. corymbosum* calluses cultivated on WPM nutrient media with the addition of phytohormones 2,4-D and BAP in different concentrations. Calluses were cultivated in the dark. Passage duration was 30 days. Results after the first passage are shown. Different letters next to the values indicate statistically significant differences (*p* < 0.05) between content of phenolic compounds in calluses when different concentrations of 2,4-D/BAP were used in nutrient media: A–C—total soluble phenolic compounds (TSPC); a–c—flavonoids; *A–D*—flavans; *a*—proanthocyanidins.

**Figure 2 plants-13-03279-f002:**
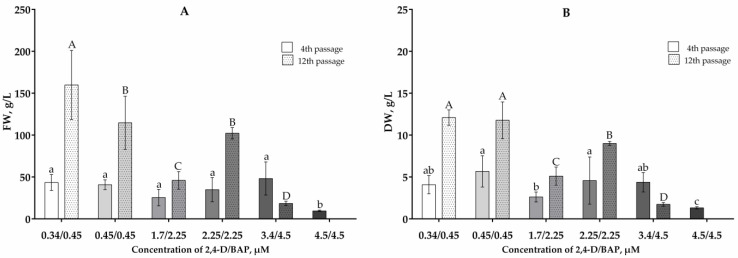
Fresh (**A**) and dry (**B**) weights of *V. corymbosum* suspension cells obtained from leaf calluses in WPM liquid nutrient media with the addition of phytohormones 2,4-D and BAP in different concentrations. The cells were cultivated in the dark. Passage duration was 30 days. Results after the 4th and the 12th passages are shown. Different letters next to the values indicate statistically significant differences (*p* < 0.05) between cell suspension weight values when different concentrations of 2,4-D/BAP were used in nutrient media: a–c—after the 4th passage; A–D—after the 12th passage.

**Figure 3 plants-13-03279-f003:**
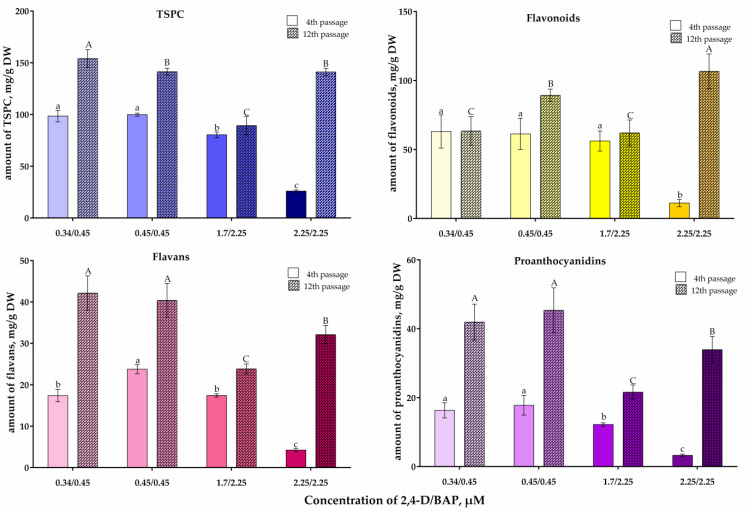
Phenolic compounds content in *V. corymbosum* suspension cells obtained from leaf calluses in WPM liquid nutrient media with the addition of phytohormones 2,4-D and BAP in various concentrations. The cells were cultivated in the dark. Passage duration was 30 days. Results after the 4th and the 12th passages are shown. Different letters next to the values indicate statistically significant differences (*p* < 0.05) between content of phenolic compounds in suspension cells when different concentrations of 2,4-D/BAP were used in nutrient media: a–c—after the fourth passage; A–C—after the 12th passage.

**Figure 4 plants-13-03279-f004:**
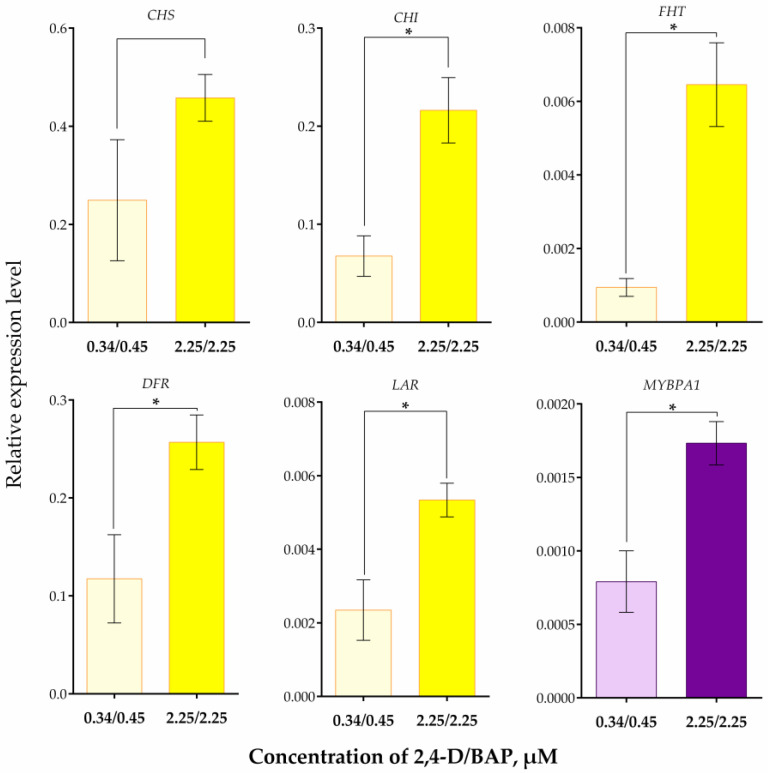
Expression of genes of flavonoid biosynthesis enzymes in *V. corymbosum* suspension cells obtained from leaf calluses in WPM liquid nutrient media with the addition of phytohormones 2,4-D and BAP in various concentrations. The cells were cultivated in the dark. Passage duration was 30 days. Results after the 12th passage are shown. Gene expression levels were normalized to three reference genes *UBQ3b*, *GAPDH*, and *EF1α*. * next to the values indicates statistically significant differences (*p* < 0.05).

**Figure 5 plants-13-03279-f005:**
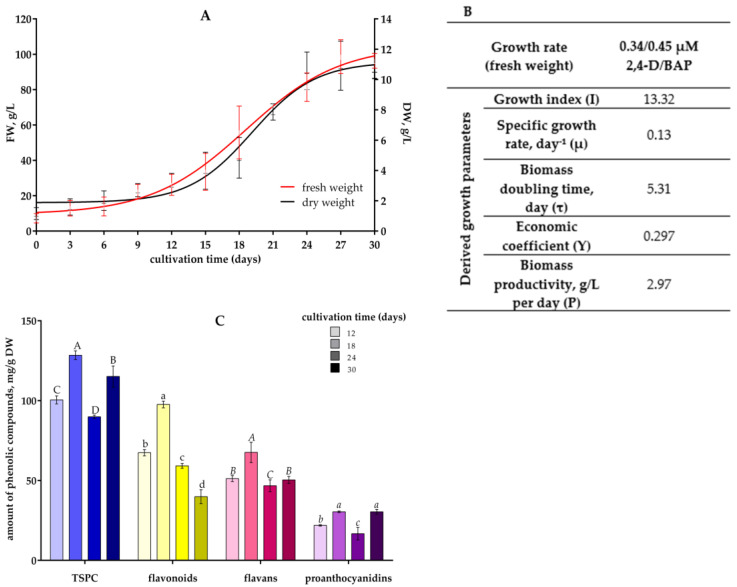
Characteristics of productivity of *V. corymbosum* cell suspension culture during one passage when cultivated in WPM liquid nutrient medium with the addition of 0.34 µM 2,4-D and 0.45 µM BAP. (**A**)—dynamics of suspension cells fresh and dry biomass accumulation. (**B**)—growth parameters calculated on the basis of the obtained growth curve of suspension cells fresh biomass accumulation. (**C**)—phenolic compounds content in suspension cells on different days of cultivation. The cells were cultivated in the dark. Passage duration was 30 days. Different letters next to the values indicate statistically significant differences (*p* < 0.05) between content of phenolic compounds in suspension cells with the addition of 0.34 µM 2,4-D and 0.45 µM BAP: A–D—TSPC; a–d—flavonoids; *A–C*—flavans; *a–c*—proanthocyanidins.

**Table 1 plants-13-03279-t001:** Characteristics of calluses obtained from leaf explants of *V. corymbosum* on WPM nutrient medium with the addition of phytohormones 2,4-D and BAP in different concentrations. Calluses were cultivated in the dark. Passage duration was 30 days. Results after the first passage are shown. Different letters next to the values indicate statistically significant differences (*p* < 0.05) between callus weight values when different concentrations of 2,4-D/BAP were used in nutrient media: A–B—fresh weight; a–b—dry weight. Bar = 1 cm.

2,4-D/BAP, µM	Morphological Characteristics	Callus Size, cm^2^	Fresh Weight of 1 Callus, g	Dry Weightof 1 Callus, g	Photo of Calluses
0.34/0.45	light, more friable, than compact	0.5–1.4	0.17 ± 0.02 A	0.018 ± 0.002 a	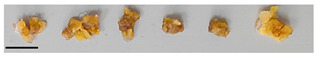
0.45/0.45	light, less often orange, more friable, than compact	0.5–1.5	0.19 ± 0.02 A	0.023 ± 0.006 a	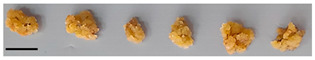
1.7/2.25	light, both friable and compact	0.5–1.3	0.15 ± 0.02 A	0.016 ± 0.002 a	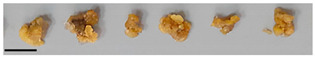
2.25/2.25	light, rarely orange, compact	0.4–1.0	0.14 ± 0.04 A	0.016 ± 0.004 a	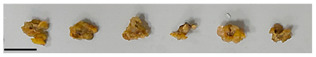
3.4/4.5	light, dark in places,compact	0.3–0.8	0.15 ± 0.04 A	0.015 ± 0.003 a	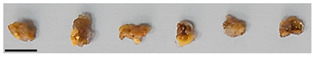
4.5/4.5	dark, some friable, some compact	0.3–0.7	0.05 ± 0.01 B	0.006 ± 0.001 b	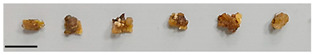
5/6.75	dark, compact	0.2–0.7	0.06 ± 0.01 B	0.007 ± 0.001 b	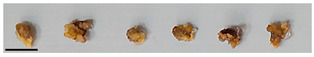
6.75/6.75	brownish-red, compact	0.2–0.5	0.04 ± 0.01 B	0.005 ± 0.001 b	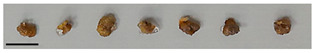

**Table 2 plants-13-03279-t002:** Characteristics of *V. corymbosum* cell suspension cultures obtained from leaf calluses in WPM liquid nutrient media with the addition of phytohormones 2,4-D and BAP in different concentrations. The cells were cultivated in the dark. Passage duration was 30 days. Results after the 4th and the 12th passages are shown. Different letters next to the values indicate statistically significant differences (*p* < 0.05) between parameters when different concentrations of 2,4-D/BAP were used in nutrient media: A–B—cell count in 1 L of medium; a–c—cell viability.

Concentration of 2,4-D/BAP, µM	0.34/0.45	0.45/0.45	1.7/2.25	2.25/2.25	3.4/4.5	4.5/4.5
Number of cells in 1 liter, million	14.9 ± 5.5 A	22.8 ± 1.4 A	11.5 ± 1.5 A	15.4 ± 4.2 A	19.1 ± 8.8 A	1.4 ± 0.4 B
Viability, %	58.8 ± 26 b	84.2 ± 2.6 a	51.0 ± 9.8 bc	62.9 ± 3 ab	63.4 ± 24 ab	30.3 ± 9.5 c
Filtered cells (4th passage)	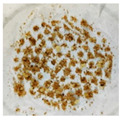	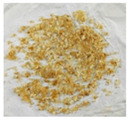	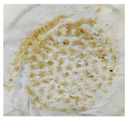	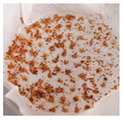	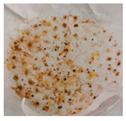	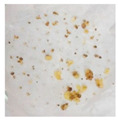
Concentration of 2,4-D/BAP, µM	0.34/0.45	0.45/0.45	1.7/2.25	2.25/2.25	3.4/4.5	4.5/4.5
Number of cells in 1 liter, million	44.8 ± 4.8 A	29.5 ± 6.7 AB	25.2 ± 9.6 B	30.4 ± 4.1 AB	19.2 ± 1.4 B	-
Viability, %	60.5 ± 8.4 b	74.1 ± 9.1 ab	63.4 ± 10.4 b	77.1 ± 5.2 a	56.6 ± 3.9 b	-
Filtered cells (12th passage)	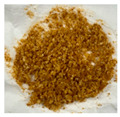	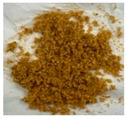	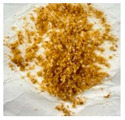	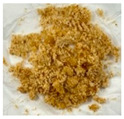	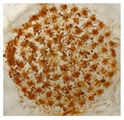	-

## Data Availability

Data are contained within the article and Appendix A.

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
