# Peer review of "Characteristics of Callus and Cell Suspension Cultures of Highbush Blueberry (Vaccinium corymbosum L.) Cultivated in the Presence of Different Concentrations of 2,4-D and BAP in a Nutrient Medium"

_plants, 2024, doi:10.3390/plants13233279_

Round 1
Reviewer 1 Report
Comments and Suggestions for Authors
This manuscript systematically investigated the effects of plant hormones 2,4-D and BAP on the biomass and phenolic compound biosynthesis in Vaccinium corymbosum callus and cell suspension cultures. The results confirmed the potential of exogenous hormones and Vaccinium corymbosum callus for the production of phenolic substances, while also provided a preliminary exploration of the biosynthetic mechanisms of phenolic compounds. This is a well-structured manuscript with comprehensive research content, appropriately designed figures, accurate result analysis, thorough discussion, and sound conclusions. I believe that with minor revisions, this manuscript is ready for publication.
1. Provide the basis for the selected concentrations of 2,4-D and BAP in the manuscript.
2. Review the significance test results for TSPC content across the treatment groups in Fig. 1, as there are some inaccuracies in the results.
3. Suggest using the 2-ΔΔCt method for calculating the relative expression levels of genes instead of the 2-ΔCt method.
4. The manuscript employs three reference genes; in the calculations presented in Fig. 4, is the average Ct value of all three reference genes used, or is it based on a specific reference gene?
Author Response
Dear Reviewer,
First of all, we would like to express our sincere appreciation for your careful attention to our manuscript and for the suggested improvements and valuable comments. We have revised the manuscript according to your remarks. Please find below the detailed answers to the questions mentioned in your review and description of the revisions.
- Provide the basis for the selected concentrations of 2,4-D and BAP in the manuscript.
Thank you very much for your comment. We have added information on the range of 2,4-D and BAP concentrations used in the Discussion section. The most commonly used concentrations of auxins and cytokinins, including 2,4-D and BAP, are from 0.1 to 5 mg/L (about 0.45–22.5 μM). For example, the best growth and production of phenolic compounds were shown for V. corymbosum calluses obtained from leaves in the presence of 0.2 mg/L 2,4-D (Ramata-Stunda et al., 2020; doi: 10.15159/AR.20.054). The maximum accumulation of anthocyanins was shown for V. ashei suspension cells, when cultivating in media with the addition of 2,4-D at concentration of no more than 0.1 mg/L (Nawa, 1993; doi: 10.1271/bbb.57.770). The maximum fresh weight of Cnidium officinale calluses was observed on a medium containing 2.3 μM 2,4-D and 2.2 μM BAP (Adil et al., 2018; doi: 10.1007/s11033-018-4340-3). The highest callusogenesis and the maximum fresh weight of Ficus religiosa calluses were observed on a medium containing 0.5 mg/L 2,4-D and 0.05 mg/L BAP (Hesami et al., 2017; doi: 10.1016/j.jgeb.2017.11.001). The fastest growth of Garcinia mangostana L. suspension cells was observed on a medium containing 2.26 μM 2,4-D and 2.22 μM BAP (Jamil et al., 2018; doi: 10.1007/s13205-018-1336-6).
In our work, we tested concentrations of 2,4-D and BAP from 0.34 to 6.75 μM. Using concentrations above 3.4 µM 2,4-D and 4.5 µM BAP, compact V. corymbosum calluses were formed, from which it was not possible to initiate cell suspension cultures. In the case of suspension cells, concentrations of 2,4-D and BAP above 2.25 µM also did not allow obtaining an actively growing culture. Therefore, we believe that using higher concentrations for V. corymbosum cell cultures is not advisable.
- Review the significance test results for TSPC content across the treatment groups in Fig. 1, as there are some inaccuracies in the results.
Thank you very much for your comment. We have made correction to Figure 1.
- Suggest using the 2-ΔΔCt method for calculating the relative expression levels of genes instead of the 2-ΔCt method.
Indeed, when assessing relative gene expression, the results of real-time PCR are most often presented as 2-ΔΔСt, thus normalizing the results of groups after a treatment or with a disease to the "control" group which is either without treatment or "healthy". However, in our case, due to the peculiarities of the technique for obtaining suspension cultures, we are talking about completely independent groups. Suspension cultures grown in the presence of 0.34/0.45 and 2.25/2.25 µM phytohormones were initiated from different calluses, also grown in the presence of different hormone concentrations, 0.34/0.45 and 2.25/2.25 µM, respectively. In this case, there is no "control" group in relation to which we could assess gene expression in these two groups. That is why we used normalization only to reference genes (2-ΔСt).
The same approach was used, for example, for assessing proanthocyanidins gene transcript abundance during blueberry fruit development and ripening (Zifkin et al. 2012; doi: 10.1104/pp.111.180950), as well as for assessing gene expression analysis on Croton lechleri callus and cell suspension of genes involved in proanthocyanidins biosynthesis (Quaggiotti et al., 2023; doi: 10.1007/s11240-023-02453-2).
- The manuscript employs three reference genes; in the calculations presented in Fig. 4, is the average Ct value of all three reference genes used, or is it based on a specific reference gene?
The values presented in Figure 4 were obtained taking into account all three reference genes listed in the Materials and Methods section and were calculated as an average. We have added corresponding explanation in the description of real-time PCR method (lines 760–761) and in Figure 4 caption (lines 254–255).

Reviewer 2 Report
Comments and Suggestions for Authors
Review of the article: "Characteristics of Callus and Cell Suspension Cultures of Highbush Blueberry (Vaccinium corymbosum L.) Cultivated in the Presence of Different Concentrations of 2,4-D and BAP in a Nutrient Medium"
The article discusses the effects of different concentrations of two phytohormones—2,4-D and BAP—on the development of callus and cell suspension cultures of highbush blueberry (Vaccinium corymbosum L.) and their ability to synthesize phenolic compounds. This study is valuable as it provides a detailed overview of the effects of phytohormones on morphology and chemical composition of in vitro cultures, which could be important for optimizing the production of plant secondary metabolites in a biotechnological context.
The study is innovative as it focuses on the relatively unexplored impact of specific concentrations of 2,4-D and BAP on phenolic biosynthesis in blueberry cultures. The introduction is well-developed, with the authors presenting a broad literature background on the role of phytohormones in plant cultures and emphasizing the particular challenges of obtaining stable, phenol-rich cell cultures. The article’s content is thus significant for research on the biotechnological production of phenolic compounds and may have practical applications.
The article is well-organized, and the results are presented clearly, making interpretation easier. The tables and graphs are well-labeled and provide strong visual support for the presented data. The descriptions of the research methods, especially the chromatographic procedures and gene expression measurement, are sufficiently detailed, facilitating comprehension of the process and replication of the study.
The research design is solid, and the results align well with the hypotheses. The use of different phytohormone concentrations allows for precise determination of their effects on various aspects of growth and biosynthesis of phenolic compounds in blueberry cultures. Particularly interesting are the results on the accumulation of phenolic compounds and the expression of flavonoid biosynthesis genes, which highlight the importance of hormonal regulation in the production of plant secondary metabolites.
Suggestions for Improvement:
- The results pertain to a specific species (Vaccinium corymbosum L.), but the authors do not discuss the possibility of applying the findings to other blueberry species or plants from the Ericaceae family. This perspective could increase the overall value of the study and the applicability of its results.
- In in vitro cell culture studies, factors such as temperature, humidity, and even light intensity can significantly influence the effects of phytohormones. It would be worth considering testing different environmental conditions to provide a fuller picture of the influence of phytohormones on phenolic biosynthesis.
- The current study focuses solely on dark cultivation with the use of 2,4-D and BAP. Including other methods, such as light cultivation or the use of other growth regulators, could expand the findings and point to alternative approaches to optimizing phenol production.
- While the article contains data on the overall phenolic content, a detailed analysis of specific phenolic compounds could add value to the work. Information on particular phenols and their known applications (e.g., antioxidant, antimicrobial) could be important for the pharmaceutical or cosmetic industry.
- The article demonstrates the impact of different phytohormone concentrations on the expression of flavonoid biosynthesis genes, but lacks a detailed discussion on potential molecular mechanisms responsible for these changes. Including considerations of signaling pathways, hormonal receptors, or specific gene regulation mechanisms could enrich the discussion.
- The article is valuable from a basic research perspective, but the authors could consider broadening the perspective to potential commercial applications of the findings. This could engage readers from industry and increase the practical significance of the study.
Comments and Questions for the Authors:
- Could the authors expand the analysis to discuss the impact of growth conditions (e.g., temperature, humidity) on the efficacy of phytohormones? Such factors can significantly influence the results.
- Do the authors plan to investigate whether similar effects can be observed in cultures of other blueberry varieties or species within the Ericaceae family? This could help generalize the findings.
- The concentrations of 2,4-D and BAP clearly affect phenolic synthesis. Would it be interesting to examine the impact of other cultivation methods (e.g., in light conditions or with other hormones) on the production of specific phenolic compounds?
- Could the authors suggest potential applications of their findings, such as in the cosmetic or pharmaceutical industries, given the high phenolic content in cell cultures?
- The authors showed that different concentrations of 2,4-D and BAP influence the expression of flavonoid biosynthesis genes. Could they expand on the potential molecular mechanisms responsible for these changes?
Author Response
Dear Reviewer,
First of all, we would like to express our appreciation for your careful attention to our manuscript and for the suggested improvements and valuable comments, which inspired further work with different ericaceous species in different cultivation conditions and with laboratory animals to test our extracts biological activities. In our laboratory, now we have callus and cell suspension cultures of V. vitis-idaea and V. macrocarpon and we would like to initiate cultures of some other species. We cultivate them both under dark and light conditions, besides, we would like to assess different light intensities effects on biomass and phenolic compounds accumulation in our cultures. Moreover, we have already transplant V. corymbosum suspension cells on media with different auxins and cytokinins. We hope that the results of all these experiments will be interesting for plant physiologists and biotechnologists, so we will be able to publish them in the nearest future in several articles.
We have revised the manuscript according to your remarks. Please find below the detailed answers to your questions.
- Could the authors expand the analysis to discuss the impact of growth conditions (e.g., temperature, humidity) on the efficacy of phytohormones? Such factors can significantly influence the results.
This article is devoted to influence of only phytohormones on accumulation of biomass and phenolic compounds by V. corymbosum calluses and suspension cells and did not involve multifactorial studies. We agree that temperature can affect the state of plant cell cultures, so our cultures were grown in closed vessels under strictly controlled temperature conditions (25°C).
- Do the authors plan to investigate whether similar effects can be observed in cultures of other blueberry varieties or species within the Ericaceae family? This could help generalize the findings.
Yes, we would like to investigate effect of hormonal composition of nutrient medium on response of other blueberry varieties or species within the Ericaceae family. At the same time, we know from many publications that different species of the same family and even different varieties of the same species can differ greatly in their response to presence of phytohormones in nutrient medium. For example, different responses to presence of phytohormones in nutrient medium was shown for calluses of different V. myrtillus lines (Botau, Bolda, 2013; http://biotechnologyjournal.usamv.ro/pdf/vol.LVI/Art8.pdf), different V. corymbosum varieties (Ramata-Stunda et al., 2020; doi: 10.15159/AR.20.054), different ericaceous species, Oxycoccus macricarpus and O. palustris (Berezina et al., 2019; doi: 10.1134/S1021443718050035), for suspension cells of different V. myrtillus lines, as well as different ericaceous species, V. vitis-idaea and Empetrum nigrum (Suvanto et al., 2017; doi: 10.1007/s00425-017-2686-8). Therefore, we cannot say that the effects we obtained are universal, and it is necessary to conduct additional research, which takes time.
- The concentrations of 2,4-D and BAP clearly affect phenolic synthesis. Would it be interesting to examine the impact of other cultivation methods (e.g., in light conditions or with other hormones) on the production of specific phenolic compounds?
This article is devoted to influence of only phytohormones on accumulation of biomass and phenolic compounds by V. corymbosum calluses and suspension cells and did not involve multifactorial studies. We agree that numerous cultivation factors, including various hormones and light, can affect synthesis of phenolic compounds. It is known that phenolic compound biosynthesis in plants is regulated by transcription factors belonging to the MYB, bHLH, WD40 (MBW) protein families. They physically interact as a complex, and such MBW complexes have been shown to be responsible for the regulation of anthocyanin, proanthocyanidin, and flavonol biosynthesis in various species and tissues, including flowers and fruits (Zifkin et al., 2012; doi:10.1104/pp.111.180950; Gu et al., 2019; doi:10.1016/j.scienta.2019.01.034). Hormone-sensitive and light-sensitive elements have been found for genes of enzymes of phenolic compounds biosynthesis. Thus, VcCHS promoter regions contain numerous cis-elements that respond to light, phytohormones, and stress factors, as well as binding sites for 36 different types of transcription factors (Wang et al., 2023; doi: 10.3390/ijms22116125). Auxin treatment can increase the expression of CHS protein and flavonoids accumulation in grape fruits (Luo et al., 2016; doi: 10.1016/j.scienta.2016.04018). This phenomenon may be due to auxin application increases ABA and CHS levels, so, increased ABA level enhances PAL and C4H gene expression, activates expression of MYB-related transcription factors, and enhances flavonoid precursor synthesis (Song et al., 2023; doi: 10.1371/journal.pone.0285134). In Arabidopsis thaliana, promoter regions in CHS, CHI, FLS, and MYB12 genes contain putative AuxREs. Moreover, these genes may contain mybRE regions. For example, 1 μM IAA supplementation induces CHS and FLS mRNA accumulation (Lewis et al., 2011; doi: 10.1104/pp.111.172502). The results of the study on effect of transcription factors on flavonoid biosynthetic pathways in blueberry fruits showed that MYBPA1, MYBA, and MYBC2 expression levels were positively correlated with anthocyanins and 3',5'-substituted polyphenols accumulation, and bHLH1 expression level was correlated with the accumulation of hydroxycinnamic acids (Günther et al., 2020; doi: 10.3389/fpls.2020.00545). In contrast, MYB4 expression level had a negative correlation, indicating a possible repression of flavonoids biosynthesis. MYB12 enhances the expression of early flavonol biosynthetic genes CHS and FLS, while anthocyanin regulators MYB75 control the expression of late DFR biosynthetic genes. Overexpression of MYB12 and MYB75 in transgenic A. thaliana plants significantly increased flavonoids accumulation with strong antioxidant activity (Wang et al., 2021; doi: 10.3390/ijms22116125). The results of the study on nectarine showed that the transcription factor MYBPA1 can control promoters of polyphenol biosynthetic genes and activate LAR transcription (Ravaglia et al., 2013; doi: 10.1186/1471-2229-13-68).
We have added information on mechanisms of regulation of phenolic compounds biosynthesis gene expression to the Discussion section.
- Could the authors suggest potential applications of their findings, such as in the cosmetic or pharmaceutical industries, given the high phenolic content in cell cultures?
We assume that callus and suspension cell cultures we obtained can subsequently be used as a source of antioxidants and become the basis for development of biologically active substances. However, we understand that, first of all, it is necessary to prove high antioxidant properties of extracts of our cell cultures and test their effect on animals. Such studies are being conducted for V. corymbosum leaf and fruit extracts. For example, V. corymbosum phenolic compounds are increasingly being considered as a promising adaptogenic and therapeutic agent for a wide range of pathological conditions, including liver damage, cardiovascular disease, neurodegenerative disorders, diabetes, obesity, etc. (Sun et al., 2019; doi: 10.1139/cjpp-2019-0031; Wang et al., 2019; doi: 10.1016/j.jff.2018.11.037; Li et al., 2020; doi: 10.1016/j.phymed.2020.153209; Sivapragasam et al., 2023; doi: 10.1016/j.tifs.2023.01.002).
We have included information on medicinal use of V. corymbosum extracts in the Introduction section.
- The authors showed that different concentrations of 2,4-D and BAP influence the expression of flavonoid biosynthesis genes. Could they expand on the potential molecular mechanisms responsible for these changes?
We have answered this question earlier (see point 3).

Reviewer 3 Report
Comments and Suggestions for Authors
This manuscript focuses on characterizing callus and cell suspension cultures of blueberries cultivated in the presence of different concentrations of 2,4-D and BAP. The manuscript provides useful information on the effect of phytohormones on plant secondary metabolite production. Before further consideration, some issues need to be clarified:
Introduction:
There are many kinds of plants that accumulate high phenolic compounds, which is why the authors chose the highbush blueberry Vaccinium corymbosum L. as the model plant in this study. Please provide more detail in the introduction.
Results:
- There are two main types of plant callus in in vitro culture: friable and compact. The generation of each type of callus depends on the ratio of phytohormones. How many types of callus were detected in this study, and what is the majority type? Please provide clear data for each treatment in Table 1 (you may add more columns separately from that of morphological characteristics).
- Are there any results on gene expression in callus culture? It is better to compare the gene expression in callus and suspension cells.
Discussion:
Please discuss why increasing the concentrations of phytohormones tends to give dense or compact calli rather than soft or friable ones.
Materials and methods:
What kind of callus is used to establish cell suspension culture, friable or compact callus?
Author Response
Dear Reviewer,
First of all, we would like to express our sincere appreciation for your careful attention to our manuscript and for the suggested improvements and valuable comments. We have thoroughly revised the manuscript according to your remarks. Please find below the detailed answers to the questions mentioned in your review.
- There are many kinds of plants that accumulate high phenolic compounds, which is why the authors chose the highbush blueberry Vaccinium corymbosum L. as the model plant in this study. Please provide more detail in the introduction.
Thank you! We have added the relevant information to the Introduction section (lines 72–79).
- There are two main types of plant callus in in vitro culture: friable and compact. The generation of each type of callus depends on the ratio of phytohormones. How many types of callus were detected in this study, and what is the majority type? Please provide clear data for each treatment in Table 1 (you may add more columns separately from that of morphological characteristics).
Thank you! We have made the appropriate changes to the text and Table 1. When using low concentrations of 2,4-D and BAP (up to 1.7/2.25 μM), we obtained predominantly friable V. corymbosum calluses. With increasing concentrations of phytohormones, calluses became compact.
- Are there any results on gene expression in callus culture? It is better to compare the gene expression in callus and suspension cells.
Indeed, we are interested in comparing gene expression in callus and suspension cells, and we have now started such studies. This is certainly interesting, but this is a separate labor-intensive and time-consuming task, so this will be the topic of a further publication. Moreover, the topic of comparing expression in two types of cultures is rather poorly covered in the scientific literature, and therefore is of particular interest for our future experiments.
Both in calluses and suspension cells, the expression of three genes DFR, ANS, and ANR selected as indicators of proanthocyanidin biosynthesis was assessed for Croton lechleri (Quaggiotti et al., 2023; doi: 10.1007/s11240-023-02453-2). It was significantly higher in dark calluses than in pale calluses and decreased to basal levels in suspension cells. Failure to accumulate metabolites in cultured cells may be due to either feedback-inhibition of key biosynthetic enzymes or non-enzymatic degradation of reaction products in the culture medium.
- Please discuss why increasing the concentrations of phytohormones tends to give dense or compact calli rather than soft or friable ones.
We assume that an increase in phytohormones concentration in nutrient medium leads to activation of plant cell wall compaction (strengthening) and lignification, which, in turn, leads to formation of compact calluses. This assumption is based on the information available in the literature on increase in ethylene synthesis under the action of auxin, which promotes cell wall lignification and strengthening. The effect has been shown, for example, for Amorphophallus muelleri (Agung et al., 2023; doi: 10.18860/elha.v9i2.26105) and Catharanthus roseus (Zavala-Ortiz et al., 2024; doi: 10.1016/j.sajb.2024.03.003) calluses. Lignification is under positive control of MYB15 transcription factor (Chezem et al., 2017; doi: 10.1105/tpc.16.00954), and it is known that members of this family of factors have hormone-sensitive elements (Wang et al., 2023; doi: 10.3390/ijms22116125).
- What kind of callus is used to establish cell suspension culture, friable or compact callus?
We have added the relevant information to the Materials and Methods section. We used all calluses (obtained on all variants of media with different concentrations of 2,4-D and BAP) to initiate suspension cells. We were able to obtain well-growing V. corymbosum cell suspension cultures only from friable calluses.

Round 2
Reviewer 2 Report
Comments and Suggestions for Authors
Thank you to the authors for their efforts in addressing the previous comments. The changes introduced have significantly improved the quality and clarity of the article. The feedback has been addressed in accordance with the guidelines, as evident in the revised manuscript.
I am pleased that the authors have incorporated suggestions regarding, for example, structure, language, and data detail, which have positively impacted the coherence and readability of the work. The current version of the article meets my expectations, and I can recommend it for publication. Thank you for the effective collaboration.